# Epistatic interactions between *PHOTOPERIOD1, CONSTANS1* and *CONSTANS2* modulate the photoperiodic response in wheat

**Lindsay M. Shaw**[1,2☯], **Chengxia Li**[1,3☯], **Daniel P. Woods**[1,3☯], **Maria A. Alvarez**[1☯], **Huiqiong Lin**[1,3], **Mei Y. Lau**[1], **Andrew Chen**[1¤], **Jorge Dubcovsky**[1,3]*

**1** Department of Plant Sciences, University of California, Davis, California, United States of America,
**2** Currently at Queensland Alliance for Agriculture and Food Innovation, The University of Queensland, Brisbane, QLD, Australia, **3** Howard Hughes Medical Institute, Chevy Chase, Maryland, United States of America

☯ These authors contributed equally to this work.
¤ Current address: School of Agriculture and Food Science, The University of Queensland, St Lucia, QLD, Australia.
* jdubcovsky@ucdavis.edu

**Data Availability Statement:** The sequence data has been deposited in GenBank with accession numbers MT043302 (CO-A1), MT043303 (CO-B1), MT043304 (CO-A2), and MT043305 (CO-B2).

## Abstract

In Arabidopsis, *CONSTANS* (*CO*) integrates light and circadian clock signals to promote flowering under long days (LD). In the grasses, a duplication generated two paralogs designated as *CONSTANS1* (*CO1*) and *CONSTANS2* (*CO2*). Here we show that in tetraploid wheat plants grown under LD, combined loss-of-function mutations in the A and B-genome homeologs of *CO1* and *CO2* (*co1 co2*) result in a small (3 d) but significant (*P*<0.0001) acceleration of heading time both in *PHOTOPERIOD1* (*PPD1*) sensitive (*Ppd-A1b*, functional ancestral allele) and insensitive (*Ppd-A1a*, functional dominant allele) backgrounds. Under short days (SD), *co1 co2* mutants headed 13 d earlier than the wild type (*P*<0.0001) in the presence of *Ppd-A1a*. However, in the presence of *Ppd-A1b*, spikes from both genotypes failed to emerge by 180 d. These results indicate that *CO1* and *CO2* operate mainly as weak heading time repressors in both LD and SD. By contrast, in *ppd1* mutants with loss-of-function mutations in both *PPD1* homeologs, the wild type *Co1* allele accelerated heading time >60 d relative to the *co1* mutant allele under LD. We detected significant genetic interactions among *CO1, CO2* and *PPD1* genes on heading time, which were reflected in complex interactions at the transcriptional and protein levels. Loss-of-function mutations in *PPD1* delayed heading more than combined *co1 co2* mutations and, more importantly, *PPD1* was able to perceive and respond to differences in photoperiod in the absence of functional *CO1* and *CO2* genes. Similarly, *CO1* was able to accelerate heading time in response to LD in the absence of a functional *PPD1*. Taken together, these results indicate that *PPD1* and *CO1* are able to respond to photoperiod in the absence of each other, and that interactions between these two photoperiod pathways at the transcriptional and protein levels are important to fine-tune the flowering response in wheat.

Mutant lines are available for distribution from multiple copies of the TILLING populations distributed worldwide, with specific locations in the John Innes Centre and the University of California Davis.

**Funding:** JD received support for this project from the National Research Initiative Competitive Grants 2017-67007-25939 and 2016-67013-24617 from the USDA National Institute of Food and Agriculture (NIFA, https://nifa.usda.gov/) and by the Howard Hughes Medical Institute (https://www.hhmi.org/). DW is a Howard Hughes Medical Institute Fellow of the Life Sciences Research Foundation (http://www.lsrf.org/). The funders had no role in study design, data collection and analysis, decision to publish, or preparation of the manuscript.

**Competing interests:** The authors have declared that no competing interests exist.

## Author summary

An understanding of the mechanisms involved in the regulation of wheat heading time is required to engineer more productive varieties better adapted to new or changing environments. A large proportion of wheat's natural variation in heading time is determined by differences in genes controlling the photoperiodic response. In this study, we show that the wheat *PHOTOPERIOD1* (*PPD1*) gene has a stronger effect on heading time than *CONSTANS1* (*CO1*) and *CO2* in the regulation of the photoperiodic response, and that complex genetic interactions among these genes are important to fine-tune heading time. Using loss-of-function mutants for both *CO1* and *CO2*, we demonstrate that these genes are not required for *PPD1* to perceive differences in photoperiod and regulate heading time. Similarly, we show that in the absence of *PPD1*, *CO1* can accelerate heading time more than 60 days in response to long days. Our results indicate that each of these two wheat photoperiod pathways can respond to differences in photoperiod even in the absence of the other one. Differences in the relative importance of these two pathways and in their epistatic interactions have contributed to the diversity of photoperiodic responses observed in different grass species.

## Introduction

A precise adjustment of flowering time to optimal environmental conditions is critical to maximize plant reproductive success. Plants have evolved complex regulatory mechanisms to adjust flowering initiation in response to seasonal changes and other environmental cues. Many plants, including the temperate cereals, require exposure to extended periods of cold (vernalization) to gain competence to flower [1]. After their vernalization requirement has been fulfilled, many plants still require inductive photoperiods to accelerate flowering. Based on their light requirements for floral induction, plants can be classified into long day (LD), short day (SD) or day-neutral plants [2].

The regulatory pathway that controls flowering in response to photoperiod is best characterized in the model plant *Arabidopsis thaliana*, a long day plant that flowers earlier when the days are longer than a critical threshold. In this species, the external coincidence model explains the integration of light signals with the circadian clock. The clock determines diurnal oscillations of the gene *CONSTANS* (*CO*), which results in the coincidence of the expression peak with the light phase under LD and with the dark phase under SD. Light stabilizes the CO protein, resulting in the induction of *FLOWERING LOCUS T* (*FT*) and the promotion of flowering under LD. Under SD, the degradation of the CO protein in the dark prevents the activation of *FT* [3–5]. Thus, *co* mutants in Arabidopsis flower later than wild type under LD but at the same time as wild type under SD [6].

CO belongs to a family of transcription factors characterized by two zinc finger B-Box domains at the N-terminus and by a CONSTANS, CONSTANS-like, TIMING OF CAB EXPRESSION 1 (CCT) domain at the C-terminus, a structure conserved in Arabidopsis and the grass homologs [7]. The *Oryza sativa* (rice) genome contains a single *CO* gene designated as *Hd1* [8], whereas two paralogs designated as *CO1* and *CO2* have been identified in *Brachypodium distachyon*, barley and wheat [9]. In the last three species, the predicted CO1 protein lacks conserved cysteine residues at the start of the second B-box, suggesting a modified or non-functional B-box 2 in the temperate grasses [7, 9]. Phylogenetic and comparative genomics analyses suggest that *CO1* and *CO2* paralogs originated by a segmental duplication at the

base of the grass family, with *CO1* located in the ancestral position colinear with rice *Hd1*, whereas *CO2* was lost in rice and is likely not functional in sorghum [9]

Overexpression of *CO1* and *CO2* under the maize *UBIQUITIN* promoter in spring barley, promoted flowering under both LD and SD conditions by upregulating *FT1* [10, 11]. However, both *Ubi*::*CO1* and *Ubi*::*CO2* transgenic lines flowered 42–43 d later in SD than in LD indicating the existence of additional genes controlling the photoperiodic response [10, 11]. In barley, the response to photoperiod is determined mainly by the pseudo-response regulator gene *PHOTOPERIOD1* (*PPD1*) [12], also known as *PRR37* in rice [13] and *Sorghum bicolor* (sorghum) [14]. This gene is regulated by the circadian clock and the phytochrome-mediated light signaling pathway and encodes a protein with a pseudo-receiver domain and a CCT domain [14, 15].

In wheat, natural deletions in the promoter region of *PPD-A1* (*Ppd-A1a* allele) and *PPD-D1* (*Ppd-D1a* allele) are associated with increased transcript levels and accelerated heading under SD relative to the ancestral *Ppd1b* allele [16, 17]. As a result, plants carrying the *Ppd1a* alleles show reduced differences in heading time between LD and SD and are designated as photoperiod insensitive (PI), whereas plants carrying the *Ppd1b* allele are designated as photoperiod sensitive (PS) [16, 17]. However, wheat PI lines still exhibit a significant acceleration of flowering under LD relative to SD and, therefore, the wheat PI designation is used in this study to indicate a reduced photoperiodic response rather than a lack of a photoperiodic response.

Under LD conditions, the wheat *PPD1* gene induces the expression of *FT1*, which encodes a mobile protein that travels to the shoot apical meristem (SAM), where it promotes the expression of the floral homeotic MADS box transcription factor *VERNALIZATION1* (*VRN1*) and the synthesis of gibberellic acid (GA) [18]. The simultaneous presence of GA and *VRN1* results in the up-regulation of *SUPPRESSOR OF OVEREXPRESSION OF CONSTANS1* (*SOC1*) and *LEAFY* (*LFY*) genes, which are required for normal and timely spike development, stem elongation and spike emergence or heading [18]. In photoperiod sensitive wheats grown under SD, the SAM transitions to the reproductive stage, but stem elongation and spike development proceed very slowly and heads fail to emerge or emerge extremely late [19]. These results indicate that in wheat, the photoperiod pathway has a larger impact on the duration of spike development and stem elongation than on the initial transition between the vegetative and reproductive stages [18, 19]. The addition of gibberellic acid (GA) to PS spring wheat grown under SD accelerates spike development, but only in lines that express *VRN1* under SD [18].

The photoperiod and vernalization pathways converge at the regulation of *FT1* in the leaves. The role of *PPD1* as a LD flowering promoter is antagonized in winter wheats by the role of *VRN2* as a LD flowering repressor [20, 21], which prevents the induction of *FT1* during the fall. Long exposures to cold temperatures during the winter (vernalization) result in the induction of *VRN1* both in the leaves and in the SAM [22]. In the leaves, the presence of *VRN1* prevents the upregulation of *VRN2* in the spring [23], which results in the upregulation of *FT1* under LD. In the SAM, the upregulation of *VRN1* initiates the transition of the SAM to the reproductive phase, but spike development proceeds slowly until the LD induction of *FT1*. The arrival of *FT1* to the SAM further up-regulates *VRN1* and induces GA accumulation, which results in accelerated spike development and stem elongation [18].

In addition to the natural *PPD1* alleles (PS and PI), induced loss-of-function mutations in all *PPD1* homeologs (henceforth, *ppd1*) have been developed for both hexaploid [24] and tetraploid wheat [19]. These *ppd1* plants show large delays in spike development and heading time under LD, and spikes usually emerge very late, fail to emerge or are aborted under SD [19, 24]. The flowering phenotypes of the wheat *ppd1* mutants are more drastic than those described for the barley photoperiod insensitive allele *ppd-H1*, which has only amino acid changes and is

likely a hypomorphic allele rather than a complete loss-of-function allele [12]. It is important to differentiate the photoperiod insensitive allele in barley that refers to delayed heading under LD, from the PI allele in wheat that refers to accelerated flowering under SD.

*CO1* and *CO2* overexpression results in barley suggest a role of these genes as flowering promoters [10, 11]. However, overexpression under a constitutive promoter results in elevated transcript levels and ectopic expression in tissues or developmental phases that are different from those of the natural alleles, which complicates the interpretation of results. The use of loss-of-function mutations can provide a better understanding of the role of these genes under more natural conditions. In this study, we investigate the role of *CO1* and *CO2* on the photoperiod response in wheat using loss-of-function mutants for both genes in the tetraploid spring wheat variety Kronos. In addition, we combine loss-of-function mutations in *co1*, *co2* and *ppd1* to study the effect of their genetic interactions on heading time and on the transcription profiles of critical flowering genes. We also investigate the interactions among the proteins encoded by these genes. Taken together, these analyses show complex genetic interactions among *CO1*, *CO2* and *PPD1* on heading time that reflect similarly complex interactions at the transcriptional and protein levels.

## Results

### Identification of *CO1* and *CO2* loss-of-function mutants in wheat

We identified a total of 153 mutations in the A and B homeologs of *CO1* and *CO2* in the Kronos tetraploid TILLING population [25, 26] using primers and PCR conditions listed in Supporting Information (S1 Table). Among the 153 mutations, 74 resulted in non-synonymous amino acid changes, and five introduced early stop codons resulting in truncated non-functional proteins (S2 Table). These five truncation mutations were validated by sequencing and one for each homeolog was selected to study the function of *CO1* and *CO2* (Fig 1). We backcrossed the individual mutant lines with wild-type Kronos at least twice to reduce background mutations.

We generated a *co1* loss-of-function mutant by crossing *co-A1* mutant T4-395 (Q242*) with *co-B1* mutant T4-1170 (Exon2 acceptor splice site mutation). To generate the *co2* loss-of-function mutant, we crossed *co-A2* mutant T4-202 (Exon 2 acceptor splice site mutation) with *co-B2* mutant T4-391 (Q193*). Finally, we crossed the *co1* and *co2* mutants and generated a *co1 co2* double mutant. The stop codon mutations in *co-A1* and *co-B2* truncate the carboxyl terminal half of the CONSTANS protein including the CCT domain (Fig 1), which is critical for protein function in both Arabidopsis and cereals [27]. Likewise, the splice-site mutations selected in the *co-B1* and *co-A2* mutants result in intron retention and premature stop codons that also eliminate the conserved CCT domain. Based on these large truncations we concluded that *co1 co2* has loss-of-function mutations for all *CO1* and *CO2* homeologs.

### Effect of *PPD1* photoperiod sensitive (PS) and insensitive (PI) alleles on heading time and spike development

We transferred the four *co1 co2* mutations generated in Kronos-PI into a near-isogenic Kronos-PS line [18] to study the effect of these mutations in two different backgrounds with large differences in heading time under SD. To establish a baseline for comparison, we first describe the effect of the two *PPD1* alleles in lines carrying the wild type *Co1* and *Co2* alleles (Fig 2A, blue bars).

Under LD, heading time was 2.5 d earlier in Kronos-PI (52.3 ± 0.4 d) than in Kronos-PS (54.8 ± 0.3 d). Although the difference was small, it was highly significant ($P < 0.0001$) and consistent with previously published results [15]. Under SD, the differences between the two

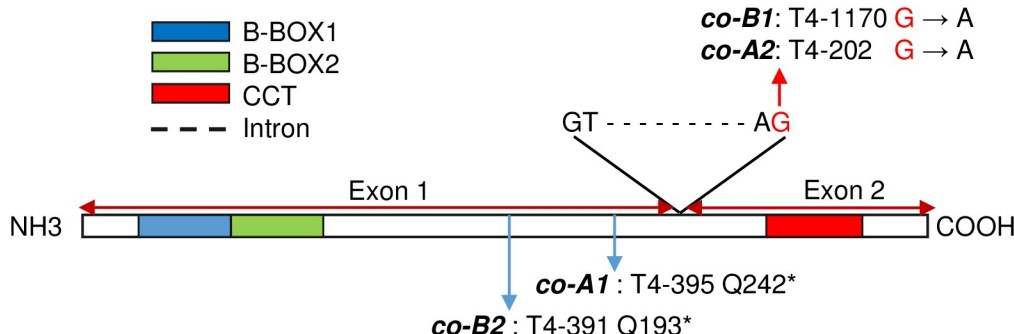

**Fig 1. *CO1* and *CO2* gene structure and position of the selected TILLING mutations.** Both *CO1* and *CO2* genes have two exons, which encode two B-BOX domains at the N-terminus and a CCT domain at the C-terminus. The selected *co-A1* and *co-B2* mutations, located in the first exon after the two B-Box domains, are predicted to generate premature stop codons, whereas the *co-B1* and *co-A2* mutants have acceptor splice-site mutations that result in intron retention and premature stop codons that eliminate the complete exon 2 including the CCT domain. The T4 numbers are the mutant identification. The schematic gene structure is based on *CO-A1* from Kronos (GenBank MT043302).

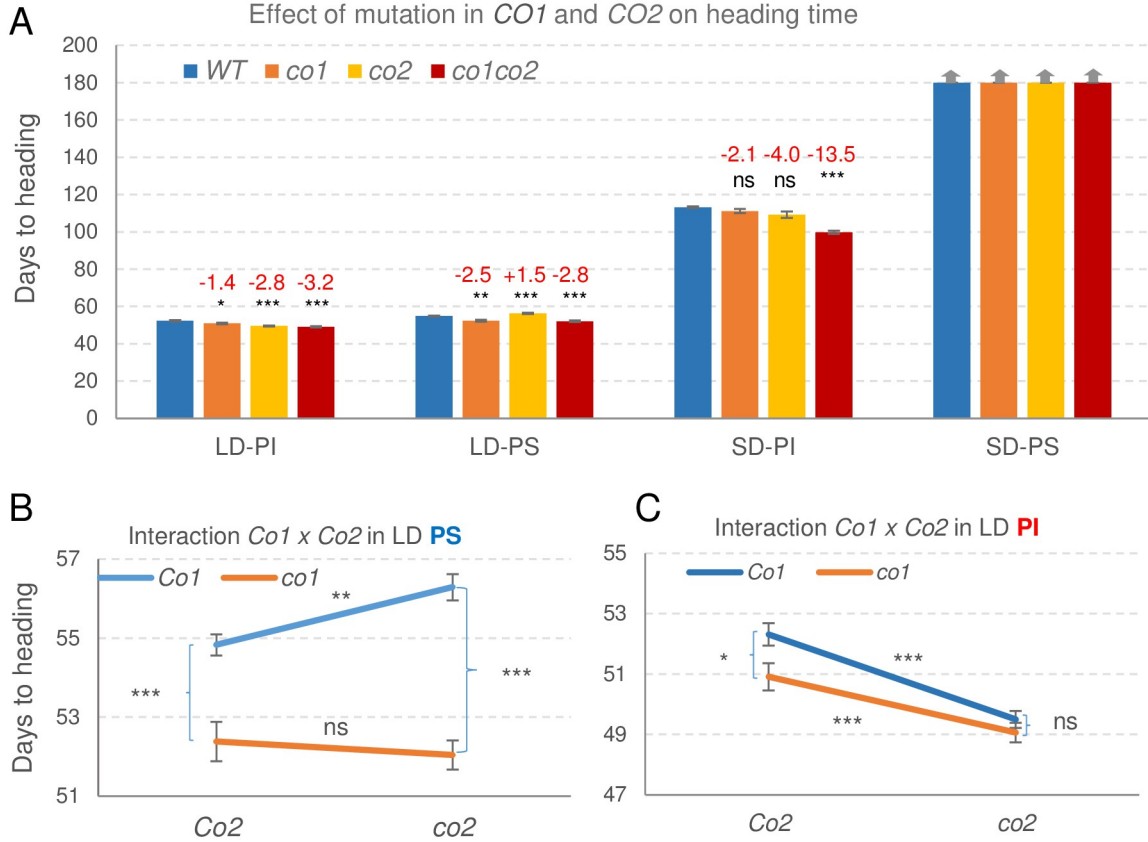

**Fig 2. Heading time of wild type, *co1*, *co2* and *co1 co2* mutants under LD and SD in Kronos-PI (*Ppd-A1a*) and Kronos-PS (*Ppd-A1b*) backgrounds.** (**A**) Averages are based on 15–19 plants in LD-PI (2 experiments), 21–24 plants in LD-PS (3 experiments) and 8 to 14 plants in SD-PI and SD-PS (1 experiment). Red numbers indicate differences in heading time relative to the wild type (mut-WT), with *P* values from a Dunnett test. Vertical arrows above the bars indicate that spikes have not emerged at 180 d (end of the experiment). (**B-C**) Interaction graphs for *CO1* and *CO2* in Kronos-PS (*Ppd-A1b*) and Kronos-PI (*Ppd-A1a*) backgrounds. The interaction was significant in B (*P* = 0.0052) but not in C (S3 Table). Error bars are standard errors of the means (henceforth, s.e.m.). * = *P* < 0.05, ** = *P* < 0.01, *** = *P* < 0.001, ns = not significant.

genotypes were much larger (>66 d, $P < 0.0001$), with Kronos-PI heading at $113.2 \pm 0.4$ d and Kronos-PS spikes failing to emerge before the experiment was terminated at 180 d. The earlier flowering of Kronos-PI under SD reduced the difference between SD and LD in this genotype (61 d) relative to Kronos-PS (>125 d, Fig 2A blue bars).

To determine if the observed differences in heading time were the result of changes in the duration of the vegetative phase or the spike development and elongation phases, we dissected SAMs of the main tillers from the different genotypes (from 4- to 14-leaf stage, S1 Fig). Under LD, the SAMs from both Kronos-PI and Kronos-PS reached the early stages of spike development by leaf four (W3-W3.25, Waddington scale [28]), the terminal spikelet stage (W3.5) by leaf six, and headed at 52–55 d (S1 Fig). Under SD, the SAMs of both genotypes were at the vegetative stage by leaf four and started to elongate by leaf six. After that point, the two genotypes diverged, while Kronos-PI transitioned to the terminal spikelet and lemma elongation stage at the eighth leaf, Kronos-PS reached the same developmental stage at leaf 12 (S1 Fig).

## Effect of *CO1* and *CO2* on heading time in the presence of functional *PPD1* alleles

Loss of *CO1* and *CO2* function resulted in small but significant changes in heading time. The *co1 co2* double mutant headed 3 d earlier than the wild type in both Kronos-PI and Kronos-PS ($P < 0.0001$) under LD, and 13.5 d earlier than the wild type in Kronos-PI under SD ($P < 0.0001$, Fig 2A). In the Kronos-PS background under SD, spikes failed to emerge or were aborted in most of the plants by the end of the experiment at 180 d, limiting our ability to determine difference on heading time between the *co1* and *co2* mutant and wild type alleles under this condition.

Even though the differences in heading time between *co1 co2* and wild type alleles were larger under SD (13.5 d) than under LD (3 d), they were still smaller than the differences between the *PPD1* photoperiod sensitive and insensitive alleles under SD (>67 d), or the differences between LD and SD in Kronos-PI (61 d) or Kronos-PS (>125, Fig 2A blue bars). These results indicate that, in wheat, the *PPD1* alleles have a stronger effect on the photoperiodic response than the *CO1* and *CO2* genes.

Except for the *co2* mutant that headed later than the wild type in the Kronos-PS background under LD (+1.5 d, $P < 0.0001$), all the other *co1* and *co2* single and double mutants headed significantly earlier than their respective wild type controls under both LD and SD (Fig 2A). These results suggest that under the tested conditions, *CO1* functions as a mild heading time repressor in all backgrounds, and *CO2* as a mild heading time promoter or repressor depending on the *PPD1* and *CO1* alleles present and the photoperiod (Fig 2A).

**Long day experiments.** To explore the interactions between *PPD1*, *CO1* and *CO2* under LD, we performed a factorial ANOVA combining data from three experiments in Kronos-PS and two in Kronos-PI, using experiments as blocks (nested within genotypes). This analysis showed highly significant effects on heading time for *PPD1* ($P < 0.0001$), *CO1* ($P < 0.0001$) and *CO2* ($P = 0.0004$) and highly significant two-way interactions for *PPD1* x *CO1* and *PPD1* x *CO2* ($P < 0.0001$, S3A Table). The *CO1* x *CO2* interaction was not significant ($P = 0.3437$) but the three-way interaction was significant ($P = 0.0046$, S3A Table), suggesting the possibility of different *CO1* x *CO2* interactions in the different *PPD1* backgrounds that cancelled each other in the combined analysis.

To test this possibility, we performed separate *CO1* x *CO2* factorial ANOVAs for Kronos-PS and Kronos-PI. In the Kronos-PS background, *CO1* showed a stronger effect ($P < 0.0001$) on heading time than *CO2* ($P = 0.1448$) and their interaction was significant ($P = 0.0052$, Fig 2B, S3B Table). By contrast, in the Kronos-PI background, *CO2* showed a stronger effect ($P <$

0.0001) than *CO1* (*P* = 0.0157) and the interaction was not significant. (*P* = 0.1934, Fig 2C, S3C Table). In summary, these results indicate complex genetic interactions between *PPD1*, *CO1* and *CO2* in the regulation of heading time under LD.

**Short day experiments.**  Under SD, the Kronos-PS lines failed to flower before the experiment was terminated (180 d), so we were only able to study the interactions between *CO1* and *CO2* in the Kronos-PI background. All four genotypes headed between 100 and 113 d, which is 51 to 61 d later than the same lines under LD. A factorial ANOVA for heading time showed significant effects for both *CO1* and *CO2* (*P* < 0.0001) and a significant interaction between the two (*P* = 0.0029, S4 Table). This interaction was associated with the larger acceleration of heading time in the double mutant (13.5 d) than in the single *co1* (2.1 d) and *co2* (4.0 d) mutants (Fig 2A). This result indicates a stronger effect of each mutation in the presence of the mutant allele of the other gene than in the presence of the wild type allele (S4 Table).

To see if the *co1* and *co2* mutations in the Kronos-PS background were associated with delays in SAM differentiation or in spike development, we dissected apices for three plants from each genotype every week from 4 to 11 weeks, and included wild type Kronos-PI as a control. Under SD, all five genotypes reached the double-ridge stage (W2.5) by week 6 (Fig 3A and 3B). From there, the Kronos-PI genotype developed more rapidly than the four Kronos-PS genotypes, which showed similar developmental rates in the wild type and the *co1* and *co2* mutants (Fig 3B). By week 11, the developing spikes in the Kronos-PI control were 3–13 cm long including awns, the peduncles were 5 to 14 mm long, and the developing flowers were at the elongated stigmatic branches stage (>W7.5). By contrast, all four Kronos-PS lines at the same time point showed no peduncle elongation, produced spikes of less than 0.5 cm, and the flowers still had the stylar canal open (<W6.0). These results indicate that the *co1* and *co2* mutations have a limited effect on spike development and elongation in the Kronos-PS background under SD compared with the strong effect of the *Ppd-A1a* in Kronos-PI.

## Effect of *CO1* and *CO2* on heading time in the absence of functional *PPD1* alleles

To investigate the effect of *CO1* and *CO2* in the absence of *PPD1*, we introgressed the *co1* and *co2* mutations into a Kronos line with loss-of-function mutations in both *PPD-A1* and *PPD-B1* (*ppd1*) [19]. As before, we describe first the differences in SAM development and heading time between Kronos-PS and *ppd1* mutants carrying the wild type *Co1* and *Co2* alleles to establish a baseline for comparison (Fig 4A, blue bars).

Under LD, the plants with the *ppd1 Co1 Co2* allele combination headed much later (115.5 ± 15.3 d) than the Kronos-PS controls carrying the functional *Ppd1b* allele (52.5 ± 0.7 d, Fig 4A), similar to results reported before [19]. The *ppd1* mutant plants showed a 79% reduction in number of grains per spike relative to the Kronos-PS lines (Fig 4B and 4C). Apex dissections showed a delayed transition to the double-ridge stage in the *ppd1* mutant (leaf 8 to 10) relative to Kronos-PS (<leaf 4, S1 Fig). Similarly, the terminal spikelet was formed in Kronos-PS by leaf 6 but was not formed in *ppd1* by leaf 14, when the experiment was terminated (S1 Fig). Under SD, spikes of both the *ppd1* mutant and Kronos-PS failed to emerge by 180 d. Dissection of the main tillers from these plants showed that spike development in *ppd1* lagged behind that of Kronos-PS (S1 Fig, leaf 12 and 14 and S2 Fig). These results indicate that *PPD1* is not essential for the initial transition of the SAM to the reproductive stage, but in its absence, spike development and stem elongation are severely compromised.

The introgression of the *co2* mutation in the *ppd1* background had a limited effect on heading time. Heading time of the double mutant *ppd1 co2* (119.1 ± 12.5 d) was only four days later than *ppd1*, and the differences were not significant suggesting a limited effect of *CO2* on

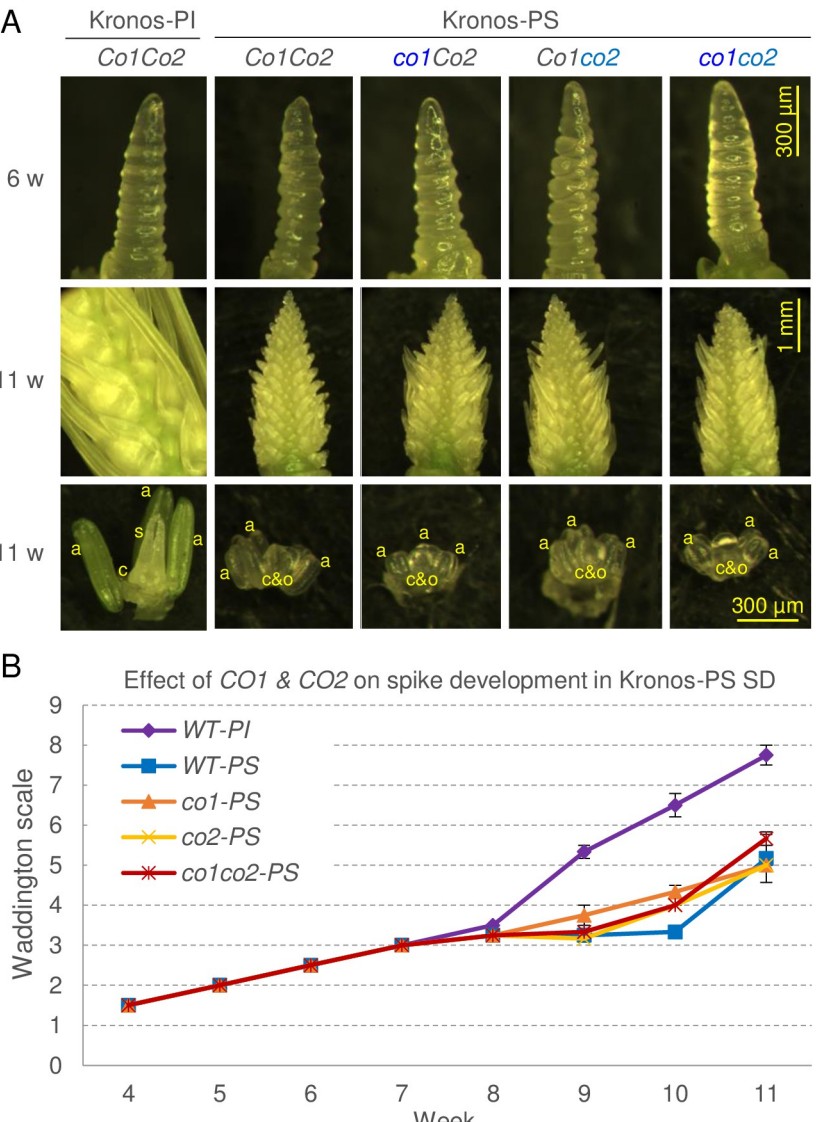

**Fig 3. Effect of *co1* and *co2* mutations on spike development in Kronos-PS under SD.** Kronos-PI was included as control. (**A**) Dissecting microscope images of developing apices at week 6 (double-ridge, W2.5) and week 11 with a detail of their floral organs. a = anthers, elongated in Kronos-PI and primordia in other genotypes, c = carpel and s = styles well developed in Kronos-PI and c&o = carpel extending around the ovule in other genotypes. (**B**) Developmental time course of SAM using the Waddington scale from 4- through 11-week-old plants (n = 3). W1 = vegetative apex, W2 = double ridge, W3 = glume primordium, W4 = stamen primordium, W5 = carpel extending round three sides of ovule, W6 = short style primordia and stylar canal open, W7 = stigmatic branches just differentiating as swollen cells on styles, W8 = stigmatic branches elongating, W9 = styles and stigmatic branches erect and stigmatic hairs differentiating.

heading time in the absence of a functional *PPD1*. The *ppd1 co2* double mutant showed an even more severe reduction in grain production than the *ppd1* mutant (89% reduction relative to Kronos-PS), but the differences were not significant.

The most important result was observed when we combined the *ppd1* and *co1* mutations. The spikes from the *ppd1 co1* plants failed to emerge before the experiment was terminated at 180 d (Fig 4A), and failed to produce any grains (Fig 4B and 4C). Even though we were not

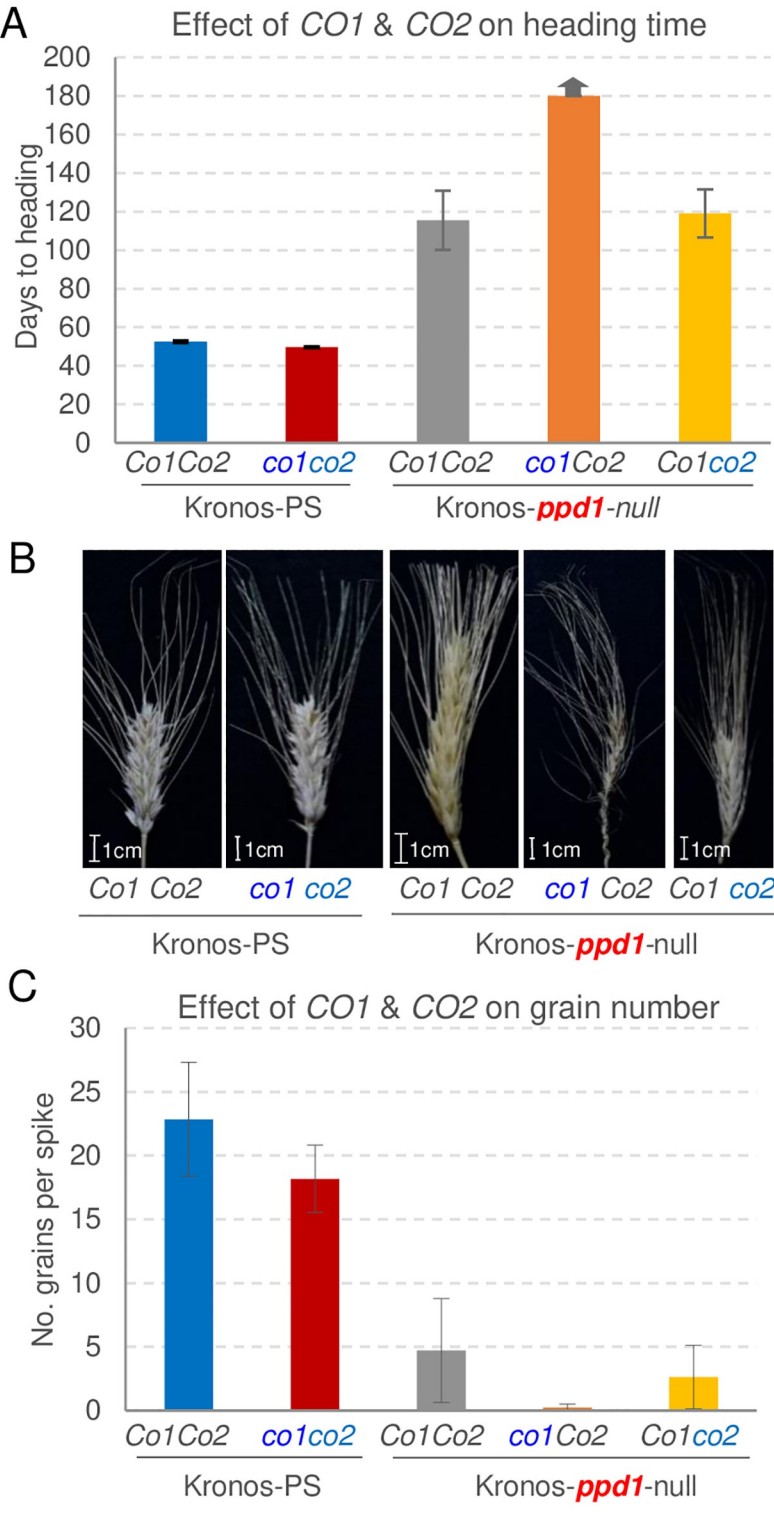

**Fig 4. Phenotypic effects of *co1* and *co2* mutations in the absence of functional *PPD1* genes under LD.** (**A**) Average days to heading in *ppd1* mutants carrying mutations in either *co1* or *co2*. Kronos-PS wild type and *co1 co2* double mutants are included as controls (arrow on top of the bar indicates heading time > 180 d). (**B**) Spike phenotypes at the end of the experiment. Floral transition and spike formation occurred in all genotypes, but in the *ppd1 co1* double mutant floral development was delayed or arrested and spikes failed to emerge before the experiment was terminated. (**C**) Number of grains per spike recovered from the same genotypes as in A. Error bars are standard errors of the means calculated from 7 to 16 plants per genotype for flowering and from 4 to 8 plants for grain number.

able to recover the triple *ppd1 co1 co2* mutant, the delayed heading time of the *ppd1 co1* mutant relative to the *ppd1* and *ppd1 co2* mutants (>60d) is sufficient to conclude that *CO1* can accelerate heading time in response to LD even in the absence of functional *PPD1* alleles.

## Effect of mutations in *PPD1*, *CO1* and *CO2* on the transcriptional profiles of flowering genes

To better understand the regulatory networks underlying the genetic interactions described above, we characterized the effect of loss-of-function mutations in *PPD1*, *CO1* and *CO2* on the transcription profiles of several flowering genes under LD and SD. Since Kronos-PI and Kronos-PS have very similar heading times under LD but not under SD, we included both genotypes in the SD experiment but only Kronos-PS under LD. To perform this experiment with a manageable number of plants, we included the *co1 co2* double mutants in both experiments but the individual *co1* and *co2* mutants only in the LD experiment We grew the plants in a growth chamber under LD or SD and sampled the fully expanded fourth leaf every 4 hours throughout a 24-h period.

Under LD, *CO1* transcripts were significantly upregulated ($P < 0.05$) in both *co2* (12 and 16 hours after the start of the light phase, Zeitgeber time ZT12 and ZT16) and *ppd1* (ZT16, ZT20 and ZT0, Fig 5A), suggesting that these two genes act as transcriptional repressors of *CO1*. The negative effect of *PPD1* on *CO1* transcription was also observed in a separate experiment using older plants (six-week-old). In leaf samples collected at ZT4 we observed a significant upregulation of *CO1* in the *ppd1* and *ppd1 co2* mutants relative to Kronos-PS (S3A Fig). No differences were detected between the *ppd1* and *ppd1 co2* mutants, which suggests that the loss of *co2* function did not affect *CO1* expression in the absence of functional *PPD1* genes (S3A Fig). Under SD, *CO1* transcripts showed a peak at ZT12 similar to LD, but that peak was in the dark phase under SD and in the light phase under LD. *CO1* was significantly downregulated in Kronos-PI during the night (ZT12 and ZT16) relative to Kronos-PS (Fig 5A).

Transcript levels of *CO2* were lower than *CO1* both under LD and SD, and showed no significant differences among most genotypes, except for a slight increase of *CO2* in the *co1* mutant under LD at ZT8 (Fig 5B) and in Kronos-PI under SD at ZT4 relative to Kronos-PS. The limited effect of the *ppd1* mutant on *CO2* expression under LD was further confirmed in six-week old plants of Kronos-PS, *ppd1* and *ppd1 co1* sampled at ZT4 in LD (S3B Fig).

Under LD, the *PPD1* gene in Kronos-PS showed a peak at ZT12 and very low expression levels at ZT0 and ZT20. The downregulation at ZT0 and ZT20 is consistent with the *PPD1* profile previously described in photoperiod sensitive *T. monococcum* [29]. The *co2* mutant showed an even higher level of *PPD1* at the ZT12 peak relative to Kronos-PS. In addition, both *co1* and *co2* mutants showed higher levels of *PPD1* than Kronos-PS at ZT4 (Fig 5C). By contrast, *PPD1* was downregulated in the *co1* and *co1 co2* mutants (ZT8 and ZT12, Fig 5C). The complex effects of the *co1* and *co2* mutations on *PPD1* transcription profiles reflect the significant two- and three-way genetic interactions among these three genes on heading time described above.

Under SD, the *PPD1* profiles were very different than under LD. Kronos-PS and Kronos-PS *co1 co2* mutant showed similar *PPD1* profiles with a peak at ZT4, but their transcript levels were lower than those of their respective Kronos-PI and Kronos-PI *co1 co2*, with the largest differences observed at dawn (Fig 5C). In addition, Kronos-PI and Kronos-PI *co1co2* showed significantly higher *PPD1* transcript levels during the night relative to Kronos-PS and Kronos-PS *co1co2*, which is consistent with previously published results in tetraploid wheat lines carrying the *Ppd-A1a* allele (PI) relative to $F_8$-near-isogenic sister lines carrying the *Ppd-A1b* allele (PS) [17].

The *FT1* profiles correlated well with heading time in the LD experiments and were almost undetectable under SD (Fig 5D). The earlier heading time of Kronos-PS *co1* and *co1 co2* under

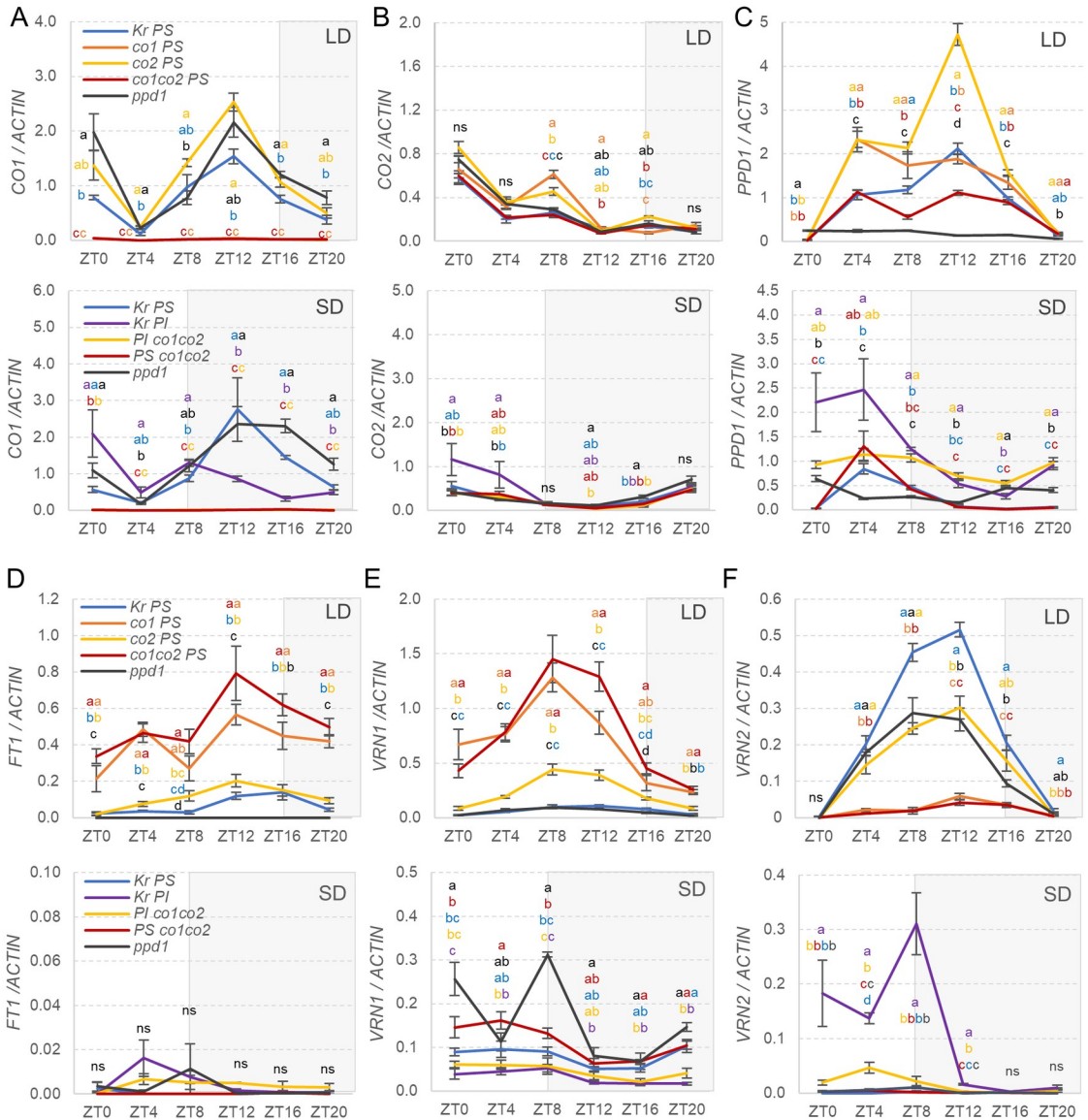

**Fig 5. Effect of loss-of-function mutations in *CO1*, *CO2* and *PPD1* on the transcriptional profiles of six wheat flowering genes in Kronos-PS and Kronos-PI (SD-only) plants grown under LD and SD.** (**A**) *CO1*, (**B**) *CO2*, (**C**) *PPD1*, (**D**) *FT1*, (**E**) *VRN1*, and (**F**) *VRN2*. LD experiments included Kronos-PS, *co1*, *co2*, *co1 co2*, and *ppd1* mutants; whereas SD experiments included Kronos-PS, Kronos-PI, and their respective *co1 co2* and *ppd1* mutants. We grew the plants in a growth chamber under 16 h light at 22 °C and 8 h darkness at 17 °C for LD, and under 8 h light and 16 h darkness for SD (same temperatures). Fully expanded 4th leaves (Zadoks scale = 14) were harvested at ZT0, ZT4, ZT8, ZT12, ZT16, and ZT20. The ZT0 value is the same as at ZT24. Shaded boxes represent night. Values are averages of four biological replicates and bars are SE of the means. Expression values are relative to *ACTIN* using the delta Ct method. Primers and primer efficiencies are listed in S5 Table (all amplify both A and B homologs of the respective genes).

LD was paralleled by significantly higher transcript levels of *FT1* throughout the day than all other genotypes (Fig 5D). Under SD, *FT1* transcript levels were slightly upregulated in Kronos-PI and Kronos-PI *co1 co2*, which were the only genotypes that headed under this condition (Fig 2).These differences were not significant in these 3-week-old plants, likely because heading in Kronos-PI under SD occurs much later (>11 weeks). We observed a significant

reduction of the transcript levels of *FT1* in the *ppd1* mutant relative to wild type, which was also consistent with its delayed heading time under LD (Fig 4).

The transcript levels of *VRN1* correlated well with the transcript levels of *FT1* under LD but not under SD (Fig 5E). Under SD at dawn and ZT8, *VRN1* transcript levels were significantly higher in *ppd1* relative to the other genotypes, and was generally higher in Kronos-PS than in Kronos-PI genotypes. In addition, *VRN1* transcripts were slightly higher in the Kronos-PS than in Kronos-PI genotypes throughout the day but the differences were statistically significant only at ZT20.

The *VRN2* locus includes two genes, but since *ZCCT1* is not functional in tetraploid wheat [30], we present only expression data for *ZCCT2*. Under LD, the transcriptional profiles of *PPD1* and *ZCCT2* were similar for the wild type Kronos-PS but not for the different mutants (Fig 5C and 5F). We observed a significant decrease of *ZCCT2* transcript levels in the *ppd1* and *co2* mutants at the ZT12 peak, and an even stronger downregulation in the *co1* and *co1 co2* mutants (Fig 5F). The interpretation of these results is complicated by the effects of the *co1* and *co2* mutations on the expression of *VRN1*, which is a known repressor of *VRN2* under LD [23] (see Discussion section). Under SD, *VRN2* was upregulated only in the Kronos-PI lines, with higher transcript levels in the presence of the functional *CO1 CO2* genes than in the Kronos-PI *co1 co2* mutants.

## Effect of *VRN1* on the transcriptional regulation of *CO1* and *CO2* under LD

To explore the potential role of *VRN1* in the transcriptional regulation of *CO1* and *CO2*, we used Kronos *vrn-A1* and *vrn-B1* single mutants and the *vrn-A1 vrn-B1* double mutant (henceforth *vrn1*) developed previously [23]. The *vrn-B1* mutant has a functional allele of *VRN-A1* that confers a dominant spring growth habit, whereas the *vrn-A1* mutant has the ancestral allele of *VRN-B1* that confers a recessive winter growth habit. The *vrn1* double mutant has a delayed heading time and a residual response to vernalization [23].

*CO1* showed significantly higher transcript levels in the late flowering *vrn-A1* and *vrn1* mutants than in the rapidly flowering *vrn-B1* mutant, both in vernalized and un-vernalized plants (Fig 6A and 6B). In all genotypes, *CO1* was downregulated to very low levels in the flag-leaf at heading time. In the *vrn1* mutant, we observed a slight decrease of *CO1* transcript levels during vernalization followed by a rapid upregulation when we moved the plants back to room temperature (RT). By contrast, in the single *vrn-A1* and *vrn-B1* mutants, which still have one functional copy of *VRN1*, *CO1* transcripts stayed low after the plants were returned to RT (Fig 6A). From previous studies, we know that *VRN1* transcript levels increase slowly during LD vernalization and then rapidly when plants are moved to RT and start to flower [23]. Taken together, these results suggest that *VRN1* acts as a transcriptional repressor of *CO1*.

*CO2* transcripts were downregulated during vernalization and showed a sharp increase when we moved the plants to RT. However, in contrast to *CO1*, this increase was observed in all three genotypes (Fig 6C and 6D). After this increase, *CO2* transcript levels remained high in the flag leaves during heading and showed no significant differences between the *vrn1* and the spring *vrn-B1* mutant (Fig 6C and 6D). These results indicate that *VRN1* has a smaller effect on the regulation of *CO2* than on the regulation of *CO1*.

## Effect of photoperiod and phytochromes on the transcriptional regulation of *CO1* and *CO2*

The strong upregulation of *CO1* in the *ppd1* mutant indicated that *PPD1* is a transcriptional repressor of *CO1* (Fig 4A). Since phytochromes *PHYC* [15] and *PHYB* [31] are both required for the LD transcriptional activation of *PPD1*, we explored the effect of mutations in these two

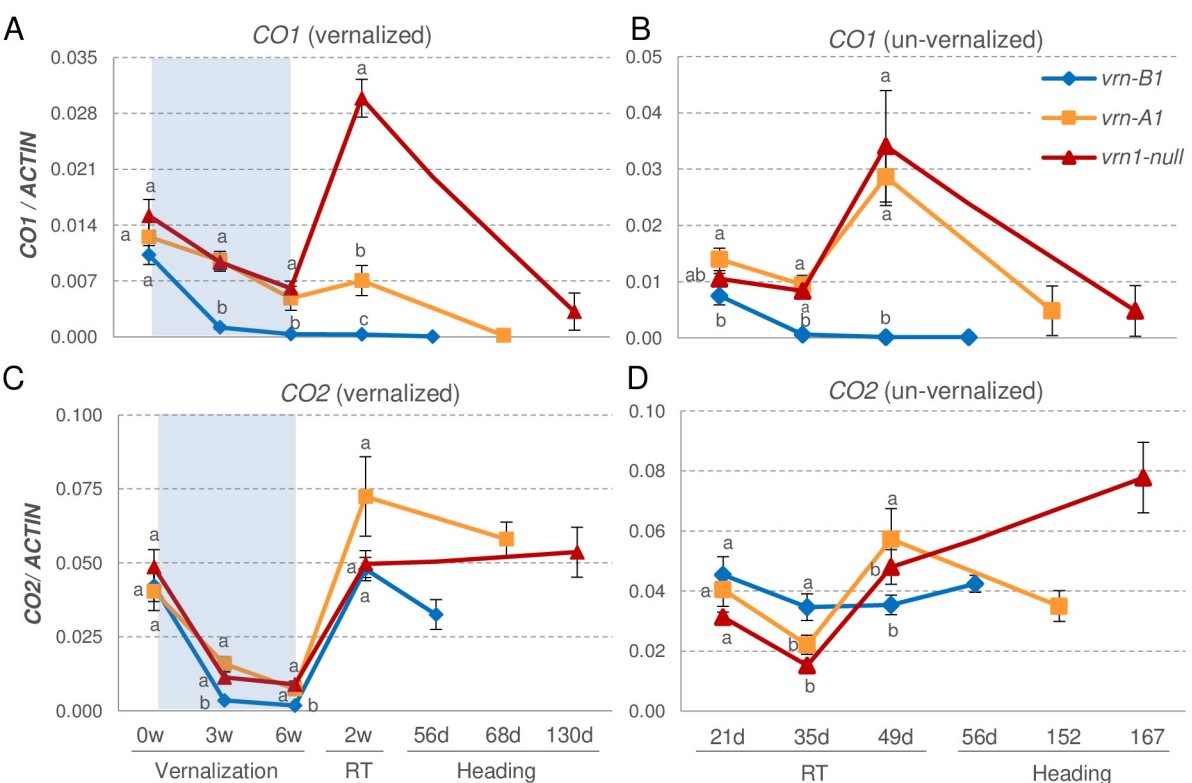

**Fig 6. Effect of mutations in Kronos *vrn-A1* (winter growth habit), *vrn-B1* (spring growth habit) and *vrn-A1 vrn-B1* (*vrn1*, very late heading) on the transcriptional profiles of *CO1* and *CO2*.** All mutants are homozygous and in the Kronos-PI background. (**A-B**) *CO1* and (**C-D**) *CO2* transcription profiles. **A** & **C**) More recently expanded leaves from three weeks old plants were sampled before vernalization (0w), at 3w and 6w vernalization (4˚C), 2 weeks after return to room temperature (RT = 22˚C day and 18˚C) and at heading time (HT), which was different for each genotype. **B** & **D**) Same genotypes but grown at RT without vernalization. Points are averages of eight plants and error bars are standard errors of the means. Means with different letters are significantly different in Tukey test for that time point (*P* < 0.05). Note that the upregulation of *CO1* after the return of the plants from vernalization occurred only in the *vrn1* mutant that has no functional copies of *VRN1*.

phytochromes on the expression profiles of *CO1* and *CO2*. To do this, we used recently published RNAseq studies for leaf samples (4th leaf, ZT 4h) from loss-of-function mutants *phyB* and *phyC* in Kronos-PI under both LD [31] and SD conditions [32].

A factorial ANOVA for transcripts per million reads (TPM) using photoperiod (LD and SD) and genotype (WT, *phyC* and *phyB*) as factors, revealed significant differences between SD and LD for all four genes (*CO-A1*, *CO-B1*, *CO-A2* and *CO-B2*, S4 Fig). We also detected difference between genotypes and genotype x photoperiod interactions for the *CO1* homeologs but not for the *CO2* homeologs. The significant interaction was associated with the LD downregulation of *CO1* homeologs in the wild type and their upregulation in the *phyB* and *phyC* mutants (S4 Fig). Both *CO2* homeologs were upregulated by LD and showed no significant differences between wild type and *phy* mutants. *CO-B2* showed significantly higher transcript levels than *CO-A2* for all genotypes (S4 Fig), whereas for *CO1* the transcript levels and the differences were larger in *CO-A1* relative to *CO-B1*.

## Protein-protein interactions among CO1, CO2, PPD1, VRN2, PHYC and PHYB

To test if the interactions among *CO1*, *CO2* and *PPD1* observed at the genetic and transcriptional levels were associated with interactions at the protein level, we performed yeast-two-

hybrid (Y2H) and split yellow fluorescent protein (YFP) assays among the three proteins. In a previous study, we reported positive Y2H interactions between the CCT domains of CO1 and CO2, but not between these two proteins and a truncated version of PPD1 including the CCT domain [33]. In this study, we used a complete PPD1 protein and detected strong Y2H interactions between PPD1 and both CO1 and CO2. The Y2H interaction between PPD1 and CO1 was stronger than between PPD1 and CO2 (Fig 7A).

We further validated the interactions between CO1 and CO2 (Fig 7B) and between CO1 and PPD1 (Fig 7C and 7D) using split-YFP assays in wheat protoplasts. We initially failed to detect the split-YFP interaction between CO1 and PPD1 in Kronos, so we used protoplasts extracted from an early flowering loss-of-function mutant for *elf3* in Kronos, which has increased levels of *PPD1* transcripts [29, 34]. We were able to detect the CO1-CO2 and CO1-PPD1 interactions (but not the CO2-PPD1) by split-YFP in the *elf3* mutant when protoplasts were exposed to light. For both CO1 and PPD1, we detected the proteins by Western blots only after we exposed the protoplasts to light but not when we incubated them in the dark after transfection, suggesting that light is required to stabilize these proteins.

Finally, we tested pairwise interactions among CO1, CO2, PPD1, VRN2, PHYC, and PHYB proteins by Y2H assays (Fig 7E and S5 Fig). In addition to interacting with CCT-domain proteins PPD1 (Fig 7A) and VRN2 [33], CO1 and CO2 showed positive interactions with both PHYC and PHYB (Fig 7E and S5 Fig) in the Y2H assays. The interaction between CO1 and PHYC was stronger than between CO1 and PHYB, whereas the interaction between CO2 and PHYB was stronger than between CO2 and PHYC. Both phytochromes showed positive interactions with VRN2 but not with PPD1 (Fig 7E and S5 Fig). In addition, we showed that the full length PPD1 protein did not interact with VRN2 in Y2H (S5 Fig). CO2, VRN2, PHYC and PHYB were able to form homodimers [15, 33] but not CO1 or PPD1 (S5 Fig). The diagrammatic representation of these interactions reveals an extensive and complex network of protein-protein interactions, with CO1 and CO2 providing a connection between the LD flowering promoter PPD1 and the LD flowering repressor VRN2 (Fig 7E).

## Discussion

### Genetic interactions among *CO1*, *CO2*, *PPD1* and *VRN2* modulate photoperiodic responses in grasses

**Long day grasses.** Natural variation in wheat heading time in response to photoperiod has been mapped mainly to the three *PPD1* homeologs [16, 17, 36]. Similarly, a genome wide association study of 220 spring barley varieties found that most of the variation in heading time (31%) was associated with the *PPD1* locus and variation in the *CO1* locus had a limited contribution (<5%) [37]. These results agree with the limited effect on heading time detected in this study in the Kronos *co1*, *co2*, and *co1 co2* mutants compared with the large differences detected between *PPD1* alleles (Fig 2) and mutants (Fig 4). Even when both *CO1* and *CO2* homeologs were replaced with truncated mutants (*co1 co2*), plants headed only three days earlier than the wild type under LD and 13 days earlier under SD (Fig 2). By contrast, loss of function mutations in *PPD1* delayed heading time approximately two months in LD (Fig 4), and differences between the *Ppd-A1a* and *Ppd-A1b* alleles were even larger under SD (Fig 2).

The effect of the *co1 co2* mutations on the acceleration of heading time was small, but highly significant ($P < 0.0001$) and consistent across experiments under LD in Kronos-PI and Kronos-PS and under SD in Kronos-PI (Fig 2). This was an unexpected result because transgenic barley plants overexpressing *CO1* [10] or *CO2* [11] flowered earlier than the wild type under both LD and SD. These transgenic results had led to the general assumption that *CO1* and *CO2* operate as flowering promoters in the temperate grasses. However, our results show that

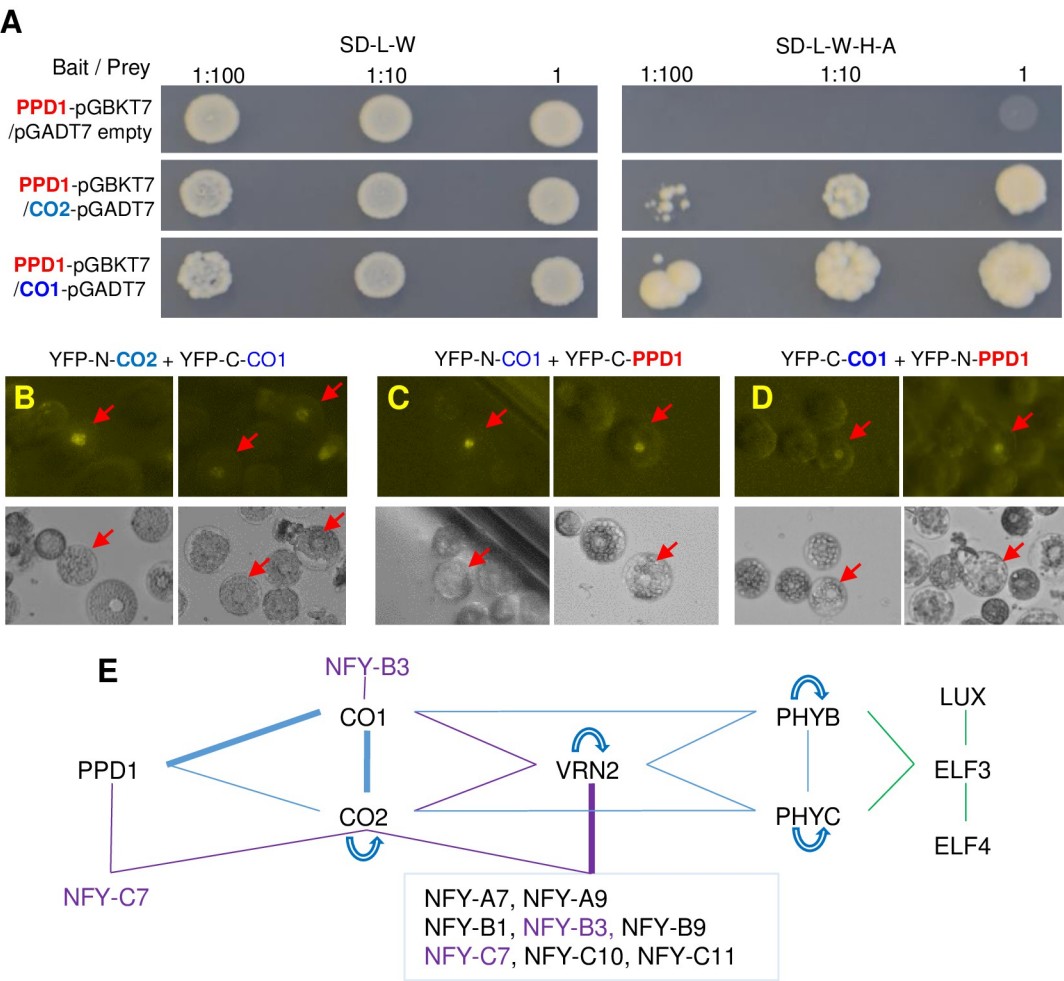

**Fig 7. Protein interactions.** (**A**) Yeast-two-hybrid (Y2H) interactions between PPD1 and CO1 or CO2. Left panel: SD medium lacking Leucine and Tryptophan (SD-L-W) to select for yeast transformants containing both bait and prey. Right panel: interaction on SD-L-W-H-A medium (no L, W, H (Histidine) and A (Adenine)). Dilution factors = 1, 1:10 and 1:100. (**B-D**) Split-YFP in wheat protoplast. (**B**) Positive nuclear interactions between CO1 and CO2. (**C-D**) Positive nuclear interactions between CO1 and PPD1 in two different YFP-N/YFP-C configurations. (**E**) Summary of Y2H interactions among wheat CO1, CO2, PPD1, VRN2, PHYB, PHYC, and ELF3 from S5 Fig. Green lines are based on published results from *Brachypodium* [35] and violet lines on published results from wheat [24]. Thick lines indicate interactions validated by split-YFP in wheat protoplasts (blue) or *in vitro* coIP (violet). Curved arrows indicate self-dimerization.

when *CO1* and *CO2* are expressed under their native promoters in their expected tissues and developmental stages (and in the presence of functional *PPD1* and *VRN2* alleles) they operate mainly as mild LD and SD flowering repressors rather than as flowering promoters (except for *CO2* in Kronos-PS under LD, Fig 2).

This inconsistency may be generated by the elevated and ectopic expression levels of *Ubi*::*CO1* and *Ubi*::*CO2* in the transgenic plants, but also by differences in the genetic backgrounds. The two transgenic barley lines were developed in the variety Golden Promise [10], which carries the *ppd-H1* allele with reduced sensitivity to LD and does not have any functional copy of *VRN2*. By contrast, Kronos has functional *PPD1* and *VRN2* genes. Our results show that in a Kronos line with no functional copies of *PPD1* (*ppd1*), *CO1* accelerates heading time under LD (Fig 4) similarly to the overexpression of *CO1* in Golden Promise. In addition, the

downregulation of *VRN2* in the Kronos *co1 co2* mutant may have contributed to the earlier heading time of these mutants (Fig 2), whereas no interactions with *VRN2* are possible in Golden Promise.

The previous results suggest the existence of complex genetic interactions among the photoperiodic genes, a hypothesis that is also supported by significant digenic interactions in the factorial ANOVA including *CO1*, *CO2* and *PPD1* (S3 Table). The three-way interaction among the three genes was also significant, which reflects the differences in digenic interactions between *CO1* and *CO2* in Kronos-PS (significant) and Kronos-PI (not significant, S3 Table). The contribution of the *PPD1* x *CO2* interaction to the total sum of squares (9%) was more than two-fold higher than the main *CO2* effect (4%), which suggests that a large part of *CO2* effect on heading time was mediated by its modulation of the *PPD1* function (S3 Table). Strong genetic interactions between *CO1* and *PPD1* on heading time have also been reported in a barley biparental population segregating for *Ubi::CO1* and the *ppd-H1* mutation [10] and in two genome wide association studies (GWAS) in spring barley [37, 38]. In both GWAS, significant effects of *CO1* on heading time were detected among the lines carrying the *ppd-H1* allele but not among the lines carrying the functional wild type *Ppd-H1* allele [37, 38]. These strong genetic interactions indicate that the effect of a particular photoperiod gene depends strongly on genetic background, a result that has also been reported in the SD grasses as described in the next section.

Although the effect of *CO1* on heading time is limited in the presence of functional *PPD1* alleles, in the *ppd1* mutant background the presence of *CO1* was required for heading under LD (Fig 4). It has been previously shown that hexaploid wheat plants with no functional *PPD1* genes (but with wild type *CO1* alleles) were able to head under LD, and that these plants express *FT1* although at lower levels [24]. These levels may be sufficient to send enough florigen to the SAM to up-regulate both *VRN1* and GA biosynthesis and the subsequent induction of *LFY* and *SOC1*, which are critical for timely spike development and stem elongation [18]. We hypothesize that the inability of the spikes to emerge in the *ppd1 co1* mutants grown under LD or in the Kronos-PS or *ppd1* mutants grown under SD, is associated with insufficient production of *FT1* in the leaves, which results in limited levels of GA, *LFY* and *SOC1* in the SAM. Addition of exogenous GA to photoperiod sensitive *T. monococcum* genotypes that express *VRN1* under SD (due to a deletion in the *VRN1* promoter) has previously been shown to partially rescue the arrest in spike development and stem elongation under SD [18].

**Short day grasses.** In most sorghum genotypes, *SbCO1* operates as a flowering promoter in SD and as a flowering repressor in LD [39] and *SbPRR37* as a flowering repressor in LD [14, 40]. By contrast, the orthologous *PPD1* gene in barley and wheat functions as a LD heading promoter [12, 16, 19] (Fig 4). In spite of the opposite roles, the genetic interactions between *PRR37/PPD1* and *CO1* showed some similarities. In a sorghum population segregating for *SbCO1* and *SbPRR37*, *SbCO1* showed a large effect on heading time among the plants carrying the non-functional *Sbprr37* allele (48% of the variation), but had no effect among the plants carrying the functional *SbPrr37* allele [41]. This is similar to the GWAS results in spring barley described above [37, 38] and consistent with the stronger effect of the *co1* mutation in the *ppd1* mutant than in the sister lines carrying functional *PPD1* alleles described in this study.

Similar to sorghum, in most rice genotypes, *OsPRR37* (= *Hd2*) works as a LD flowering repressor [13] and *OsCO1* (= *Hd1*) as both a LD flowering repressor and a SD flowering promoter. Interestingly, transformation of a rice line deficient in *OsCO1* function with a wheat *CO2* genomic clone accelerated flowering in SD and delayed flowering in LD, complementing the *OsCO1* function [42]. Since our results indicate that wheat *CO2* functions as a SD flowering repressor (Fig 2A), the rice result suggests that the function of the wheat CO2 protein was altered when it was expressed in rice. The opposite change was observed when a late-flowering

Arabidopsis *prr7* mutant was transformed with rice *OsPRR37* driven by the Arabidopsis *PRR7* promoter. When expressed in Arabidopsis, *OsPRR37* functioned as a LD promoter of flowering in spite of its normal role as a LD repressor of flowering in rice. Based on these results, it is tempting to speculate that the different roles of *PRR37/PPD1* and *CO2* in SD and LD grasses may be the result of differences in the balance of multiple epistatic interactions among the genes involved in the photoperiodic response, rather than a difference in the specific function of their encoded proteins.

This hypothesis is also supported by changes in gene function reported in sorghum and rice plants carrying loss-of-function mutations in both *PRR37* (*prr37*) and *GRAIN NUMBER, PLANT HEIGHT, AND HEADING DATE7* (*ghd7*, a *VRN2* ortholog [43]). In the *prr37 ghd7* sorghum and rice mutants, *CO1* function changed from a flowering repressor to a flowering promoter in LD, while maintaining its usual role as a flowering promoter in SD [41, 44, 45]. In addition, in most rice genotypes carrying the *ghd7* mutation *OsPRR37* function changes from a SD promoter to a SD repressor of heading, while maintaining its role as a LD flowering repressor [46]. Genetic interactions between *VRN2* (*GHD7*) and both *CO1* (*Hd1*) [47] and *CO2* [11] have been reported also in barley, suggesting that similar mechanisms may exist in the temperate cereals. Taken together, these results indicate that genetic interactions between *PPD1/PRR37*, *VRN2/GHD7* and *CO1/CO2* play a central role in the regulation of heading time in response to photoperiod in both LD- and SD-grasses and that, therefore, the effect of particular genes may vary in different genetic backgrounds.

## Complex interactions at the transcriptional level contribute to the observed genetic interactions on heading time

The significant upregulation of *CO1* in the *ppd1* mutant under LD (Fig 5A and S3A Fig) suggests that *PPD1* may function as a transcriptional repressor of *CO1* under this condition. This hypothesis is supported by the upregulation of *CO1* in the *phyB* and *phyC* mutants (S4 Fig), where *PPD1* transcription is strongly downregulated [15, 31], and by the negative correlation between *PPD1* and *CO1* transcript levels in both *ppd1* triple mutants of hexaploid wheat 'Paragon' under LD [24], and photoperiod sensitive and insensitive isogenic lines of Paragon under SD [48]. Under SD, however, the *CO1* transcription profile in the *ppd1* mutant was similar to the Kronos-PS (Fig 5A). *CO1* transcript levels were significantly downregulated during the night in Kronos-PI relative to Kronos-PS, possibly as a result of the higher transcript levels of *PPD1* in Kronos-PI (Fig 5C). These results suggest that *PPD1* may work as a repressor of *CO1* expression under both LD and SD, but during different parts of the day (Fig 5A).

Since *PPD1* induces the expression of *VRN1* under LD (Fig 5E), and *VRN1* represses *CO1* transcription (Fig 6), we cannot rule out the possibility that the effect of *PPD1* on *CO1* transcription under LD is mediated, at least in part, by *VRN1*. When Kronos *vrn-A1*, *vrn-B1* and *vrn1* mutants were returned to room temperature after vernalization, *CO1* was rapidly upregulated in the leaves of the *vrn1* mutant but not in the other three genotypes with functional *VRN1* copies (Fig 6). An identical expression profile has been described before for *VRN2* in the same mutants [23], indicating that *VRN1* functions as a LD transcriptional repressor of both *CO1* and *VRN2*.

*CO2* transcript levels were not significantly affected by the *ppd1* mutant allele under LD or SD (Fig 5B) or by the loss-of-function mutants of *VRN1* in a LD experiment (Fig 6C and 6D). Similarly, no changes in *CO2* transcription were observed between Paragon-PI and Paragon-PS lines grown under SD [48], between Kronos-PI and the *phyB* or *phyC* mutants (S4 Fig), or between Kronos-PS and *ppd1* mutants (S3B Fig). However, mutations in *co2* affected the LD diurnal transcriptional profiles of *VRN1* and *VRN2* (Fig 5E and 5F). Under LD, *CO1* transcript

levels were 64% higher in the *co2* mutant than in the Kronos-PS control at the ZT12 peak ($P < 0.05$, Fig 5A), suggesting that CO2 functions as a negative transcriptional regulator of *CO1*. This is also supported by the downregulation of *CO1* transcripts in a barley line overexpressing *CO2* [11]. The effect of *CO2* on *CO1* and *PPD1* transcription may contribute to the significant genetic interaction on heading time observed between *CO1*, *CO2* and *PPD1* in this study (S3B Table).

Overexpression of *CO1* and *CO2* in winter barley was associated with the upregulation of *VRN2*, both under LD and SD, where *VRN2* is not normally expressed [11]. A similar upregulation of *VRN2* has been reported in *Brachypodium* plants overexpressing *CO1* under LD [49]. Consistent with this result, knock-down of *CO1* expression in *Brachypodium* [49] or loss-of-function mutants in *co1*, *co2*, or *co1 co2* in Kronos in our study (Fig 5F) were associated with the down regulation of *VRN2* transcripts in the mutants relative to Kronos-PS under LD, and in Kronos-PI *co1 co2* relative to Kronos-PI under SD. The regulation of *VRN2* by CO1 may be direct, as it was recently shown in rice that a complex formed by Hd1 (CO1), GHD8 and OsHAP5b can bind the promoter of *GHD7* (*VRN2*) and activate its transcription [50]. However, we cannot rule out indirect effects because CO1 may act as a transcriptional repressor of *VRN1* in LD (Fig 5E), and *VRN1* is a LD transcriptional repressor of *VRN2* [23, 51]. Taken together, these results are consistent with *CO1* and *CO2* being promoters of *VRN2* transcription in the temperate grasses, a role that is likely more important during the fall than in the spring, when high expression of *VRN1* results in the downregulation of both *CO1* and *VRN2*.

Photoperiod and vernalization genes converge on the regulation of *FT1*, whose transcript levels are well correlated with heading time [11, 52]. Here we also observed a good correlation between *FT1* and *VRN1* transcript levels and heading time under LD (Figs 2 and 5D). Since *PPD1* is a critical LD transcriptional promoter of *FT1* [12, 35], it was not surprising to see almost undetectable levels of *FT1* in the *ppd1* mutant, which were correlated with low *VRN1* transcript levels and late heading time under LD (Fig 5D and 5E).

It has been previously shown that deletions in a CArG box located in the *VRN1* promoter in photoperiod sensitive *T. monococcum* are associated with increased *VRN1* transcript levels under SD, suggesting the existence of a SD repression mechanism of *VRN1* transcription [20]. The upregulation of *VRN1* in the *ppd1* mutant under SD (Fig 5E) suggests the possibility that *PPD1* may be involved in this SD repression mechanism. A strong SD upregulation of *VRN1* has also been reported in *phyC* and *phyB* mutants of Kronos-PI, which also showed a significant down-regulation of *PPD1* [32].

Under LD, the reduced transcript levels of *VRN2* in the *ppd1* mutant relative to Kronos-PS indicates that *PPD1* contributes to the expression of *VRN2* under LD (Fig 5F). A similar contribution seems to occur under SD, where *VRN2* was highly upregulated in Kronos-PI and to a lesser extent in Kronos-PI *co1 co2* (Fig 5F), the two genotypes that show a significant increase in *PPD1* transcripts under SD (Fig 5C). Since a short pulse of light in the middle of the night induces *PPD1* transcription and is sufficient to greatly accelerate Kronos-PS flowering in SD [19], we speculate that the higher transcript levels of *PPD1* during the night in the two Kronos-PI lines may be important for the acceleration of heading time under SD. The positive role of *PPD1* on the upregulation of *VRN2* under SD has also been reported in hexaploid wheat lines carrying the photoperiod insensitive *Ppd-D1a* allele but not in sister lines carrying the photoperiod sensitive *Ppd-D1b* allele [54], and in young winter barley plants [11].

To summarize the dynamics of these transcriptional interactions during the seasons, we propose the following working model for the ancestral photoperiod sensitive winter wheats. In the fall, *PPD1*, *CO1* and *CO2* promote the expression of *VRN2* in the leaves, which prevents flowering by repressing the transcriptional induction of *FT1*. During the winter, the increase in *VRN1* expression in the apex [22] promotes the initial stage of reproductive development

whereas its upregulation in the leaves results in the gradual downregulation of both *VRN2* and *CO1*, which facilitates the promotion of *FT1* by *PPD1* and the acceleration of spike development and stem elongation in the spring.

## Protein-protein interactions add an additional level of regulation to wheat heading time

In addition to the complex transcriptional interactions described above, the proteins encoded by the photoperiod genes form an intricate network of protein-protein interactions that contribute an additional layer of complexity to the photoperiodic regulation of heading time. In Arabidopsis, the central photoperiod gene *CO* is regulated at the transcriptional and posttranscriptional level. A complex transcriptional regulation is required to establish the presence of *CO* mRNA during the evening of a long day, when light can stabilize the CO protein, resulting in the activation of *FT* transcription [27].

Several PRR proteins promote flowering in Arabidopsis by interacting with and stabilizing the CO protein during the day specifically under LD, and indirectly through their roles in the circadian clock [55]. Mutations in *PRR* genes delay Arabidopsis flowering under LD but not under SD (similar to *PPD1* in wheat), with double mutants showing a stronger delay [56]. The late-flowering Arabidopsis *prr7* mutant can be complemented by rice *OsPRR37* [57], suggesting some conserved functions in rice and Arabidopsis. In this study, we detected interactions between wheat PPD1 full-length protein and both CO1 and CO2 in Y2H and validated the CO1-PPD1 interactions by split-YFP (Fig 7). These results suggest that the interactions between CO and PRRs likely precede the split between monocot and dicot plants.

In spite of these similarities, there are also important differences between the grass *PPD1/PRR37* gene and the homologous *PRR* genes from Arabidopsis. The Arabidopsis *PRR* genes are part of the circadian clock and their mutants affect the expression of circadian clock genes and their multiple downstream targets. By contrast, the natural mutation in the barley photoperiod insensitive *ppd-H1* allele did not affect diurnal or circadian cycling of barley clock genes [58]. This result suggests that after the duplication that generated *PRR37* and *PRR73* in the grasses, which is independent of the Arabidopsis duplication that originated *PRR3* and *PRR7*, *PRR37/PPD1* acquired a more specialized role in the photoperiodic response with a more limited role in the circadian clock.

In addition to its interactions with the LD flowering promoter PPD1, CO1 and CO2 can form heterodimers with the LD flowering repressor VRN2, which is also a member of the CCT domain protein family [33]. We hypothesize that the interactions with VRN2 might be relevant to *CO1* and *CO2* function because loss-of-function mutation in *ghd7/vrn2* result in reversals of functional in *OsPRR37* (*Hd2*) [46] and *OsCO1* (*Hd1*) in rice [44, 45], and *SbCO1* in sorghum [41]. In rice, GHD7 binds to the transcription-activating domain of OsCO1, probably weakening or blocking the transcriptional activation of *Ehd1* and *Hd3a* [45]. We have initiated crosses among Kronos lines with mutations in *vrn2* [30], *co1*, *co2*, and *ppd1* to test these interactions in wheat.

Recent studies in rice showed that another CCT-domain protein, the NUCLEAR TRANSCRIPTION FACTOR Y subunit B-11 (NF-YB11 = HEME ACTIVATOR PROTEIN or HAP-3H = *Ghd8*) has large effects on heading time and significant genetic interactions with *OsGHD7*, *OsCO1* and *OsPRR37* [46]. We have previously shown that in addition to interacting with each other, CO2 and VRN2 interact with the same subset of eight NUCLEAR FACTOR-Y (NF-Y) transcription factors (Fig 7E) [33], suggesting an additional functional connection between these proteins. This connection through the NF-Y proteins seems to be less extensive between CO1 and VRN2, where only one shared NF-Y interactor has been identified so far

(Fig 7E) [33]. Members of the NF-Y family have been shown to be involved in the regulation of flowering through interactions with CCT domain proteins CO and COL in other plant species [59–61]. Using yeast-three-hybrid (Y3H) assays, we previously showed that VRN2 competes with CO2 for interactions with the NF-Ys, and that mutations in three conserved arginine (R) residues within the CCT domain of VRN2 reduce the strength of the protein interactions and competition with CO2 [33]. Since these same mutations abolish or reduce VRN2 function as a LD flowering repressor, we hypothesize that the protein-protein interactions involving VRN2-CO2-NF-Y CCT domain could be relevant for their function [33].

Phytochromes PHYB and PHYC are involved in the transcriptional regulation by light of both *PPD1/PRR37* and *VRN2* in wheat [15, 31], *Brachypodium* [62], rice [63], and sorghum [40]. In addition, OsPHYA, OsPHYB and OsGI proteins have been shown to interact directly with OsGHD7, regulating protein stability [64]. We observed similar interactions between VRN2 and both PHYB and PHYC in wheat. These two phytochromes also showed Y2H interactions with CO1 and CO2, but not PPD1 (Fig 7E and S5 Fig). Additional experiments will be required to test if these phytochrome interactions can stabilize the VRN2, CO1 and CO2 proteins in wheat as they stabilize GHD7 in rice [64] or CO in Arabidopsis [65].

In summary, we found that CO1 and CO2 proteins can physically interact with each other and with both PPD1 (a central LD flowering promoter) and VRN2 (a central LD flowering repressor), providing a physical link between these central photoperiodic proteins (Fig 7E). Among the 15 pairwise interactions tested among PHYB, PHYC, PPD1, CO1, CO2, and VRN2 80% showed positive interactions in Y2H assays indicating a complex and dense network of protein interactions. We hypothesize that this network of protein interactions provides an additional layer of molecular mechanisms that result in the complex genetic interactions in heading time detected in this study.

## Two interacting photoperiod pathways in grasses

A central conclusion of our study is that *PPD1* can perceive the differences in photoperiod and adjust heading time accordingly even in the absence of any functional copy of *CO1* or *CO2*. Similarly, *CO1* can accelerate heading time under LD in the absence of a functional copy of *PPD1*. This is also true in rice, where heading time differences between SD and LD are observed for *OsCO1* alleles in *Osprr37* mutants, and for *OsPRR37* alleles in rice *Osco1* mutants [46]. These results suggest that grasses have either two photoperiod pathways, each of them capable of perceiving the length of the day/night and of regulating heading time, or a single point of perception of the differences in photoperiod upstream of *PPD1* and *CO1*, which can then transmit the signal to both genes. Recently, it has been suggested that the dark reversion of the Pfr form of PHYC may be involved in measuring night length (molecular hourglass model), by modulating the accumulation of the ELF3 protein, a direct transcriptional repressor of *PPD1* [35]. This mechanism is different from the external coincidence model proposed for Arabidopsis [66]. Although we favour the hypothesis of separate mechanisms for *PPD1* and *CO1* perception of photoperiod, we currently cannot rule out the possibility that, in the temperate grasses, ELF3 mediates both the *PPD1* and *CO1* responses to photoperiod.

Another important conclusion from our study and from previous studies in sorghum and rice [40, 44, 46], is that the *CO* and *PRR37/PPD1* photoperiod pathways are extensively interconnected. Changes in the balance of these interactions seem to play a central role in the diversity of photoperiod responses in grasses, as suggested by changes in function observed when photoperiodic genes are swapped between LD and SD species or when the same genes are expressed in different mutant backgrounds in its own species. The different expression profiles of the photoperiodic genes in LD and SD also show that these interactions are dynamic and

affected by photoperiod. We hypothesize that the complex feedback transcriptional regulatory loops and transcriptional interactions among photoperiod genes along with the intricate interactions among their encoded proteins contribute to the complex genetic interactions in heading time observed in this study and to the diversity of photoperiodic responses in grasses.

## Materials and methods

### Plant materials

The materials used in this study included the tetraploid wheat variety Kronos (*Triticum turgidum* ssp. *durum*), which carries a functional and dominant *Vrn-A1* allele for spring growth habit, a functional but recessive *vrn-B1* allele for winter growth habit, a functional *Vrn-B2* long day repressor (*ZCCT-B2a* and *ZCCT-B2b* [30]), and the *Ppd-A1a* (photoperiod insensitive or PI) allele that confers earlier flowering under short days resulting in a reduced photoperiodic response. We also used in our study near-isogenic lines of Kronos carrying either the photoperiod sensitive *Ppd-A1b* (PS) allele [18] or no functional alleles in the *PPD-A1* and *PPD-B1* homeologs (*ppd1*) [19].

Using previously published wheat sequences for *CO1* [48] and *CO2* [42], and the Kronos transcriptome [67] we identified the A and B coding sequences for *CO1* and *CO2* in Kronos and deposited them in GenBank under accession numbers MT043302 (*CO-A1*), MT043303 (*CO-B1*), MT043304 (*CO-A2*), MT043305 (*CO-B2*). We used those sequences to identify loss-of-function mutations in *CO1* and *CO2* in the Kronos TILLING population [25, 26], which are described in detail in the Results section. We then combined mutations in the A and B homeologs from each gene to generate loss-of-function *co1* (henceforth *co1*) and *co2* mutants (henceforth *co2*). We backcrossed the individual mutant lines with wild-type Kronos at least two times to reduce background mutations and then combined all four homozygous mutations to produce a *co1 co2* mutant (henceforth *co1 co2*). In addition to the wild type Kronos-PI, the four homozygous *Co1 Co2*, *co1 Co2*, *Co1 co2* and *co1 co2* allelic classes were generated in Kronos-PS and the *ppd1* mutant. We were not able to recover the triple loss-of-function mutant *ppd1 co1 co2* from any of the tested progenies.

### Phenotyping

We stratified wheat grains at 4 ˚C for 2 d in the dark and after germination we planted them in pots in the soil. We evaluated the mutant and control lines in CONVIRON growth chambers with metal halide bulbs supplemented by florescent bulbs set to 22 ˚C during daylight and 17 ˚C during dark periods. Lights were set to 260–300 μmol m$^{-2}$ s$^{-1}$ and were on for 16 h in LD experiments and 8 h in short day (SD) experiments. We recorded heading time as the total time from planting in soil to full emergence of the spike from the sheath. Spikes were threshed and total grain number counted per spike.

### Statistical analysis

We compared heading time in different genotypes using one-way ANOVAs for single locus and factorial ANOVAs for two or more loci. We tested homogeneity of variances using the Levene's test and normality of residuals with the Shapiro–Wilks test. We transformed the data when necessary to satisfy the ANOVA assumptions. We performed all statistical analyses using SAS 9.4 (SAS Institute). Error bars in all graphs are standard errors of the means (s.e.m.).

## Gene expression analysis

To study the effect of *ppd1*, *co1* and *co2* mutant alleles on the transcription profiles of genes affecting wheat flowering, we collected leaves from plants grown in a growth chamber under LD using the same settings as described in the previous phenotypic section. For the time course experiment, we collected samples in liquid nitrogen every 4 h throughout the day from the newly expanded fourth leaf. In this experiment, we included the *ppd1* mutant and all four possible combinations of the *CO1* and *CO2* wild type and mutant alleles in the Kronos-PS (*Ppd-A1b*) background. In the SD experiment, we included Kronos-PS and Kronos-PI with their respective *co1 co2* mutants, and the *ppd1* mutant.

RNA was isolated from leaves using the Spectrum Plant Total RNA Kit (Sigma-Aldrich) following the manufacturer's recommendations. We treated the RNA samples with DNase (Promega) according to the manufacturer's protocol prior to first-strand cDNA synthesis. We assessed RNA integrity by a combination of A260/A280 and A260/A230 measurements and running samples on a denaturing formaldehyde-agarose gel as recommended in the Qiagen RNAeasy Mini Handbook. We used only samples with 260/280 and 260/230 ratios > 1.7 and no degradation. We synthesized cDNA using 2.0 µg of total RNA per sample and M-MLV Reverse Transcriptase (Promega) and poly-T primers (Sigma-Aldrich).

We performed the quantitative reverse transcription-PCR (qRT-PCR) assays using Sybr-Green Takara Master Mix in a 7500 Fast Real-Time PCR system (Applied Biosystems). For PCR, we used one cycle at 95˚C for 30 s and 40 cycles of 95˚C for 5 s, 60˚C for 5 s, and 72˚C for 34 s, followed by a melting curve program. We quantified expression using the delta Ct method ($2^{-\Delta CT}$) with *ACTIN* as endogenous control [15]. Primer efficiencies were all > 95% and their sequences are summarized in S5 Table. Primer sequences were previously described for *FT1* and *VRN1* [51], *PPD1* [48], *CO1* and *CO2* [15], and *VRN2* [30].

## Yeast-two-hybrid (Y2H) and split yellow fluorescent protein (YFP) assays

The full-length coding region of PPD1 was amplified from Kronos and the PHYB truncations were amplified from *Triticum monococcum* with primers listed in S5 Table. The amplified fragments were cloned into yeast vectors pGBKT7 and pGADT7 (Clontech, http://www.clontech.com/), which were used to transform yeast strain Y2HGold (Clontech) using the lithium acetate method. Transformants were selected on SD medium lacking leucine (L) and tryptophan (W) plates and re-plated on SD medium lacking L, W, histidine (H) and adenine (A) to test the interactions. Yeast vectors for VRN2 (full-length, both bait and prey), CO1 (prey, including both B-boxes and CCT domain, amino acids 10 to 379) and CO2 (prey, including both B-boxes and CCT domain, 15 to 356 amino acids) were previously described in [33], and those for PHYC (N-PHYC, C-PHYC and FL-PHYC) and full-length PHYB (as prey) in [15]. CO1 and CO2 showed strong auto-activation when used as baits so they were only used as preys. The full-length PHYB gave strong autoactivation when used as bait, so we generated two additional truncations N-PHYB and C-PHYB (primers in S5 Table). Only N-PHYB can be used as bait without autoactivation, and all three (N-PHYB, C-PHYB and FL-PHYB) can be used as preys in Y2H assays. The truncated versions of PHYC include amino acids 1–600 (N-terminal) and 601–1139 (C-terminal) [15], and those for PHYB amino acids (1–625 (N-terminal) and 626–1166 (C-terminal).

For the bi-molecular fluorescent complementation (BiFC or split YFP) assays, each set of wheat proteins was recombined into modified Gateway-compatible pY736 and pY735 which contain the *UBIQUITIN* promoter to generate YFP-N-terminal fragment and YFP-C-terminal-fragment fusion proteins. Wheat protoplasts were prepared, transfected and visualized as described in [68]

## Supporting information

**S1 Fig. Shoot apical meristem (SAM) and spike development.** Kronos-PI (*Ppd-A1a*), Kronos-PS (*Ppd-A1b*) and Kronos-*ppd1* loss-of-function mutant plants grown under LD (16 h light / 8 h darkness, top) and SD (8 h light / 16 h darkness, bottom). Bar is 200 μm in all figures. Samples are aligned by developmental stage (leaf number), but chronological time of dissections differed between LD and SD. Main tillers were dissected from three plants per genotype/time point and SAMs were photographed, but only one representative SAM of the three is included in the figure.
(PDF)

**S2 Fig. Dissection of developing spikes.** (**A**) Kronos-*ppd1* null mutant and (**B**) Kronos-PS control plants grown under SD and dissected 140 days (20 weeks) after sowing. Note the faster development of Kronos-PS relative to Kronos-*ppd1*-null. In both genotypes spikes failed to emerge before 180 days when the experiment was terminated.
(PDF)

**S3 Fig. Effect of *ppd1*, *co1* and *co2* loss-of-function mutations and photoperiod on *CO1* and *CO2* transcript levels.** RNA samples were collected at ZT4 from leaves of six-week-old Kronos-PS and *ppd1* plants (with and without *co1* and *co2*). (**A**) *CO1* transcript levels. (**B**) *CO2* transcript levels. Dunnett's tests were used to compare the two mutants with the wild type. Transcript levels are expressed relative to *ACTIN* using the ΔCt method. Averages and standard errors of the means were calculated from a minimum of five biological replicates per genotype.
(PDF)

**S4 Fig. Effect of *phyB*-null and *phyC*-null mutations in Kronos-PI on the transcriptional profiles of *CO1* and *CO2* under SD and LD.** Results extracted from a published RNAseq study [32]. Samples were collected from the newest expanded leaves at ZT4 from 4w-old plants under LD (n = 4) and 8w-old plants under SD (n = 8) to synchronize wild type genotypes grown under different photoperiods to a similar early reproductive stage (W3 early spike development without terminal spikelet). Comparisons between LD and SD should be interpreted with caution, since photoperiod effects and chronological time effects are conflated. Factorial ANOVAS were performed separately for each gene and P values for photoperiod (SD vs LD), genotype (WT, *phyB*, *phyC*) and their interactions are indicated below the gene names.
(PDF)

**S5 Fig. Yeast-two-hybrid (Y2H) assays.** Primers for cloning *PPD1* (from Kronos) and PHYB truncations (from *T. monococcum*) are listed in Supplementary Table S5. Primers and vectors for *CO1*, *CO2* and *VRN2* are described in [33] and those for *PHYC* and full-length *PHYB* in [15]. Transformants were selected on SD medium lacking leucine (L) and tryptophan (W) plates and re-plated on SD medium lacking L, W, histidine (H) and adenine (A) to test the interactions. Due to auto-activation, CO1 and CO2 can only be used as preys. Only N-PHYB can be used as bait without autoactivation, so this is the only PHYB clone tested for interactions with CO1 and CO2. For VRN2, we used the functional ZCCT1 paralog from T. monococcum. The PPD1-bait construct used in assays presented in this figure is the same as in Fig 7A showing a positive interaction with both CO1 and CO2.
(PDF)

**S1 Table. Genome-specific primer sequences and PCR conditions for TILLING.** We sequenced genome-specific PCR products the first time to confirm amplification of the correct

target.
(PDF)

**S2 Table. Number of mutations detected in the targeted regions of wheat *CO1* and *CO2* homologs in the Kronos TILLING population.**
(PDF)

**S3 Table. Analysis of variance for heading time under long days (LD, 16 h light / 8 h darkness).** This factorial ANOVA combined all four classes of *CO1* and *CO2* wild type and mutant alleles in photoperiod sensitive (PS, *Ppd-A1b*, three experiments) and photoperiod insensitive backgrounds (PI, *Ppd-A1a*, two experiments). For the statistical analyses, we used experiments as blocks nested within *PPD1* classes.
(PDF)

**S4 Table. Analysis of variance for heading time under short days (SD, 8 h light / 16 h darkness).** All four allelic combinations for *CO1* and *CO2* wild type and mutant alleles were analyzed in a photoperiod insensitive background (Kronos-PI).
(PDF)

**S5 Table. Primers used in the qRT-PCR and Y2H experiments.**
(PDF)

## Acknowledgments

We thank Emily Quach for her excellent technical support during this project.

## Author Contributions

**Conceptualization:** Jorge Dubcovsky.

**Formal analysis:** Jorge Dubcovsky.

**Funding acquisition:** Jorge Dubcovsky.

**Investigation:** Lindsay M. Shaw, Chengxia Li, Daniel P. Woods, Maria A. Alvarez, Huiqiong Lin, Mei Y. Lau, Andrew Chen.

**Project administration:** Jorge Dubcovsky.

**Resources:** Andrew Chen, Jorge Dubcovsky.

**Supervision:** Lindsay M. Shaw, Chengxia Li, Jorge Dubcovsky.

**Visualization:** Lindsay M. Shaw, Chengxia Li, Daniel P. Woods, Jorge Dubcovsky.

**Writing – original draft:** Lindsay M. Shaw.

**Writing – review & editing:** Lindsay M. Shaw, Chengxia Li, Daniel P. Woods, Maria A. Alvarez, Huiqiong Lin, Andrew Chen, Jorge Dubcovsky.

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
