## [Decision Letter · Decision Letter 0]

18 May 2020

Dear Dr Dubcovsky,

Thank you very much for submitting your Research Article entitled 'Epistatic interactions between PHOTOPERIOD-1, CONSTANS 1 and CONSTANS 2 modulate the photoperiodic response in wheat' to PLOS Genetics. Your manuscript was fully evaluated at the editorial level and by independent peer reviewers. The reviewers appreciated the attention to an important topic but identified some aspects of the manuscript that should be improved.

We therefore ask you to modify the manuscript according to the review recommendations before we can consider your manuscript for acceptance. Your revisions should address the specific points made by each reviewer.

[LINK]

Yours sincerely,

Sarah Hake

Associate Editor

PLOS Genetics

Gregory P. Copenhaver

Editor-in-Chief

PLOS Genetics

In addition to responding to the reviewers comments, I have a couple comments.

in reading the abstract, I was confused by the nomenclature. The mention of the Ppd1-A1a and Ppd1-A1b backgrounds, then later a mention of ppd1 null. In reading the text, I gather that PpdA1a is a promoter deletion of PPD1 and PpdA1b is ancestral, but it left me puzzled. Perhaps you can guide your naive readers through this in the abstract.

Also you mention Co-1 and CO1 as the wild-type alleles in the abstract (line 36 and 37). that makes it confusing too.

I appreciate seeing the inflorescences in figure 3, but the detail of the floral organs is lost without some indication of what we are looking at. Is it simply that the stamens are larger in the WT-PI? Also the use of Waddington's scale is useless to anyone outside of the wheat community I imagine.

In Figure 4 B, the co1;ppd1 double. What is in the photo? It looks like a spike. As you write, it did not emerge and failed to make grains, but I would say this genotype did flower. Perhaps heading time refers to the emergence of the panicle? I

Reviewer's Responses to Questions

**Comments to the Authors:**

Reviewer #1: The manuscript by Shaw et al. studies loss of function mutations of PPD1, CO1 and CO2, among others, and their effect on flowering time in wheat. The authors nicely reveal complex genetic and protein-protein interactions among those and other components of the vernalization and photoperiod response pathways. The results are solid and well organized. The Discussion introduces new hypothesis about the genetic control of flowering time in winter cereals.

I would like to suggest some minor points that can be revised in the manuscript.

Abstract, line 36 – Specify ‘wild-type Co1’ accelerated heading time …

Figure 1 is a schematic representation of the CO1 gene structure. The gene has two exons, but the figure illustrates a single model, indicating the position of the identified mutations. I guess that stop codon mutations co-A1 and co-B2 are both in exon 1 (line 161), whereas the splice-site mutations co-A2 and co-B1 correspond to the intron, causing intro retention.

Line 200 – 'The interaction was' instead of 'the interactions was'?

Lines 761-762 - I understand that the experiment was carried out in CONVIRON growth chambers as the one described in lines 774-776. Unify both sentences within the phenotyping section.

Reviewer #2: This manuscript investigates the role of CONSTANS1 (CO1) and CO2 in photoperiodic control of flowering in tetraploid wheat. co1 and co2 mutant alleles are combined with several different mutant alleles of PHOTOPERIOD1 (PPD1) to understand how these genes interact under long day (LD) and short day (SD) photoperiods. The authors show weak repressive roles for CO1 and CO2, with CO1 being the more important of the two. PPD1 has a comparatively greater role in activating flowering with or without CO1 and CO2. Importantly, CO1 promotes flowering under long days (LD) without functional PPD1, while PPD1 promotes flowering without CO1 or CO2. Gene expression analysis shows complex changes in expression of key flowering time genes that are essentially in line with observed flowering phenotypes. Complex interactions are also demonstrated upon testing of the protein-protein interactions made between CO, PPD1, VRN2, and phytochromes B and C. Because the authors how CO1/2 and PPD1 are functionally independent, this work raises interesting questions about upstream photoperiod perception pathways, as well as how these independent pathways interact downstream. This work also clearly highlights the crucial contribution genetic background makes to the activity of either CO or PPD1, which could be determined by combined transcriptional regulation and a large network of protein-protein interactions.

The work is well done, carefully interpreted, and logically presented with clearly written text.

While the Discussion is primarily focused on events occurring in the leaf to promote FT1 expression (summarized by the working model in lines 640-645), how do these genetic/protein interactions contribute to promotion of spike development in the post-floral transition SAM? This seems like an important consideration, given the authors present in the Introduction that “..in wheat, the photoperiod pathway has a larger impact on the duration of spike development and stem elongation than on the initial transition between the vegetative and reproductive stages” (lines 121-123). Furthermore, the authors show that all Kronos-PI & Kronos-PS genotypes [including co1-PS, co2-PS, and co1 co2-PS] achieved the same post floral transition stage by 6 weeks under SD (Figure 3) before development of Kronos-PI diverged from the others. They also conclude that “PPD1 is not essential for the initial transition of the SAM to the reproductive stage, but in its absence, spike development and stem elongation are severely compromised” (lines 297-299). In total, these observations indicate PPD1 and CO1/2 activity have important contributions to spike development, which presumably occurs following the action of FT1, but how is not explored in the Discussion.

For readers outside the temperate grass flowering field, there is too little introduction to the roles of VRN1 and VRN2 to follow the arguments made in the Discussion. The authors could add a few sentences in the Introduction or Discussion to clarify.

It is not clear what is meant by the inclusion of ELF3 in this context: “..photoperiod upstream of PPD1 and CO1 (e.g. ELF3)..” (line 723). It the proposal that ELF3 is the common upstream component of photoperiod perception the regulates PPD1 and CO1/2?

**Have all data underlying the figures and results presented in the manuscript been provided?**

Reviewer #1: Yes

Reviewer #2: Yes

PLOS authors have the option to publish the peer review history of their article (what does this mean?). If published, this will include your full peer review and any attached files.

Reviewer #1: No

Reviewer #2: Yes: Frank G. Harmon

---

## [Decision Letter · Decision Letter 1]

12 Jun 2020

Dear Dr Dubcovsky,

We are pleased to inform you that your manuscript entitled "Epistatic interactions between PHOTOPERIOD-1, CONSTANS 1 and CONSTANS 2 modulate the photoperiodic response in wheat" has been editorially accepted for publication in PLOS Genetics. Congratulations!

Please note that Reviewer #1 has a couple minor comments that you should attend to as you prepare the final draft of the manuscript for the production team (the editorial team will not need to re-evaluate).

Yours sincerely,

Sarah Hake

Associate Editor

PLOS Genetics

Gregory P. Copenhaver

Editor-in-Chief

PLOS Genetics

Comments from the reviewers (if applicable):

Reviewer's Responses to Questions

**Comments to the Authors:**

Reviewer #1: The authors have addressed the questions raised during the previous review. They added new sentences clarifying different aspects of the manuscript, mainly in the Introduction and Discussion sections.

Two minor issues or typos should be revised:

Lines 107-108 – Check the designation of the PPD1 alleles, Ppd1-A1a or Ppd-D1a?

Lines 296-297 –Figure 3 legend, check the Waddington scale ratings, “W7 stigmatic branches elongated and W8 sigmatic branches elongating”, is it correct?

Reviewer #2: The authors have fully addressed the issues raised in my initial review.

**Have all data underlying the figures and results presented in the manuscript been provided?**

Reviewer #1: Yes

Reviewer #2: Yes

PLOS authors have the option to publish the peer review history of their article (what does this mean?). If published, this will include your full peer review and any attached files.

Reviewer #1: No

Reviewer #2: No

**Data Deposition**

http://datadryad.org/submit?journalID=pgenetics&manu=PGENETICS-D-20-00637R1

**Press Queries**

---

## [Editor Report · Acceptance letter]

7 Jul 2020

PGENETICS-D-20-00637R1 

Epistatic interactions between PHOTOPERIOD1, CONSTANS1 and CONSTANS2 modulate the photoperiodic response in wheat 

Dear Dr Dubcovsky, 

We are pleased to inform you that your manuscript entitled "Epistatic interactions between PHOTOPERIOD1, CONSTANS1 and CONSTANS2 modulate the photoperiodic response in wheat" has been formally accepted for publication in PLOS Genetics! Your manuscript is now with our production department and you will be notified of the publication date in due course.

With kind regards,

Jason Norris

PLOS Genetics

On behalf of:
